# Beyond Average Leaderboards:
# When Explicit Graph Priors Help Tabular Foundation Models

Franck Le[1]   Keith Grueneberg[1]   Erich M. Nahum[1]   Vadim Sheinin[1]

## Abstract

Tabular foundation models (TFMs) now rival strong tree ensembles, yet most encode each row independently, ignoring relationships between instances. A natural fix is an explicit graph prior linking related rows—but does it help, and when? We study LATTICE, a lightweight, backbone-agnostic augmentation that connects rows through shared feature values and refines a TFM's frozen row embeddings with a small graph network. Across 452 classification datasets and three backbones of increasing strength (CARTE, FT-Transformer, TabPFN v3), the benefit tracks encoder weakness: the graph adds +7.8 macro-F1 points on average to the weakest encoder, +2.4 to the middle one, and essentially nothing to state-of-the-art TabPFN (−0.1). Yet this average misleads: on identifiable regimes—tables with weak label–feature signal and high class imbalance—the graph lifts even TabPFN by a median of +6.6 and up to +11.4 points. Because where it helps is predictable from cheap meta-features, a rule that augments only when the graph is expected to help beats both always- and never-augmenting on every backbone. Explicit graph priors are thus targeted, not universal, upgrades—worthwhile even atop a state-of-the-art model.

## 1. Introduction

Average benchmark leaderboards can hide an important fact about tabular learning: the best model often depends on the dataset. Recent tabular foundation models (TFMs)—including TabPFN (Hollmann et al., 2023; Grinsztajn et al., 2026), CARTE (Kim et al., 2024), and TabICL (Qu et al., 2025)—have largely closed the long-standing gap between deep networks and gradient-boosted trees. Yet most TFMs still process each row independently at inference time. This leaves out a plausible inductive bias: many tables contain repeated feature-value patterns—shared categories and recurring numeric ranges—so rows with similar local structure may also behave similarly. An explicit graph over instances is a natural way to encode this bias.

Graph priors clearly help sometimes: prior graph-augmented tabular models deliver real gains on many datasets, even if the size of the improvement varies from one table to the next (Guo et al., 2021; Yan et al., 2023; Kim et al., 2024). The practical question is therefore not whether a graph prior helps in general, but when it helps—and on which backbone. Answering it means moving past a single average ranking to pinpoint the dataset conditions under which the prior helps, and to flag those conditions in advance.

We study this question with LATTICE (Link-Augmented Tabular Transformer for Instance Connectivity Exploitation), a graph augmentation that preserves a pretrained backbone and refines its frozen row embeddings by message passing on a bipartite instance–feature-value graph (Figure 1). Because LATTICE only reads a backbone's embeddings, the same augmentation applies to any TFM. This lets us hold the graph fixed and vary the backbone, and thereby ask: how does the value of a graph prior change as the underlying encoder gets stronger? We instantiate LATTICE on three backbones of increasing strength—CARTE, FT-Transformer (Gorishniy et al., 2021), and TabPFN v3—and compare against XGBoost and CatBoost, over 452 datasets and five seeds.

Our findings reduce to three points.

1. The graph is a powerful equalizer. It adds large, highly significant gains to the weakest encoder (CARTE, +7.8 points on average) and moderate

[1]IBM, Yorktown Heights, NY, USA. Correspondence to: Franck Le <fle@us.ibm.com>.

Proceedings of the $2^{nd}$ ICML Workshop on Foundation Models for Structured Data, Seoul, South Korea. 2026.

gains to FT-Transformer ($+2.4$), pulling both up toward the performance ceiling set by state-of-the-art TabPFN. On TabPFN's own average the effect is neutral ($-0.1$)—as expected once an encoder already sits at that ceiling.

2. **Where it helps is predictable.** Even for TabPFN, cheap meta-features (label–feature predictability, class balance, fraction of continuous features) sharply separate where the graph helps from where it hurts, carving out interpretable 1–3 dimensional regimes in which it lifts even this state-of-the-art model by up to $+11$ points—and flagging the smaller regions where it should be skipped.

3. **Prediction enables routing.** A simple rule that augments a table only when the graph is predicted to help beats both always- and never-augmenting on every backbone, and the discovered patterns replicate on held-out datasets far above chance.

Together these results reframe the question from "does a graph prior help?" to "for which encoder and which datasets?"

## 2. Related Work

Graph-augmented tabular models connect features or instances to improve prediction—T2G-Former organizes features into relation graphs, multiplex GNNs build multi-view instance graphs, and CARTE encodes each row as a small graph (Yan et al., 2023; Guo et al., 2021; Kim et al., 2024)—while text-classification graphs such as TextGCN and BertGCN inspired our feature-value construction (Yao et al., 2019; Lin et al., 2021). This line of work reports mixed benefits and, crucially, evaluates a single backbone, so it cannot separate the value of the prior from the strength of the encoder it is attached to. Our contribution is to hold the graph fixed and vary the backbone—isolating when, and on which encoder, an explicit prior is beneficial—and to identify the dataset regimes, characterized by a few cheap meta-features, in which it helps or hurts.

## 3. LATTICE

LATTICE adds explicit connectivity among rows while preserving a pretrained encoder. It has two parts: (i) a backbone that maps each row to an embedding, and (ii) a graph head that refines those embeddings by message passing over an instance–anchor graph. Crucially, the backbone is frozen: LATTICE only reads its row embeddings, so any TFM can serve as the backbone.

**Backbone.** Given a dataset $\mathcal{D} = \{(x_i, y_i)\}_{i=1}^{N}$, the backbone maps each row $x_i$ to $h_i \in \mathbb{R}^d$. We use three

backbones of increasing strength: CARTE (Kim et al., 2024), FT-Transformer (Gorishniy et al., 2021), and TabPFN v3 (Grinsztajn et al., 2026).

**Instance–anchor graph.** Rather than a dense instance–instance or learned $k$NN graph, LATTICE builds a bipartite graph

$$G = (V_{\text{inst}} \cup V_{\text{anch}}, E),$$

where each anchor node is a categorical feature value or a discretized numeric bin, and $(i, a) \in E$ if row $i$ takes value $a$. Two rows interact only when they share a feature value, so connectivity reflects local feature similarity rather than a learned metric—a construction inspired by text-classification graphs (Yao et al., 2019; Lin et al., 2021). Instance nodes are initialized with the frozen embeddings $h_i$ and anchor nodes with fixed random vectors; a graph transformer layer (Shi et al., 2021) then refines the instance embeddings into $\tilde{h}_i$, which feed a linear classifier. Only the graph head is trained—the backbone and the anchor initialization are left untouched.

**Transductive inference.** We study the transductive setting: unlabeled test rows join the graph with their labels hidden, so they exchange information with training rows through shared anchors. An inductive variant, which attaches each test row to a fixed training graph at prediction time, is a natural extension we leave to future work.

## 4. Experimental Setup

**Datasets.** We use public tabular classification datasets pooled from TabArena, TabZilla, PMLB, and the TP-BERTa suite (Erickson et al., 2025; McElfresh et al., 2024; Olson et al., 2017; Yan et al., 2024). After deduplication the pool contains 452 datasets. Benchmark datasets are disjoint from backbone pretraining corpora, limiting contamination. For the cross-method leaderboard we use the 413-dataset complete-case subset on which every method returns valid results.

**Methods and protocol.** We compare native (no-graph) and LATTICE-augmented versions of each backbone against XGBoost and CatBoost (Chen & Guestrin, 2016; Prokhorenkova et al., 2018). Trees and graph heads are tuned with Optuna; TabPFN v3 uses its official defaults. Every method is run with five random seeds under fixed train/validation/test splits, and we summarize a dataset by the median macro-F1 across seeds. For each backbone and dataset we define

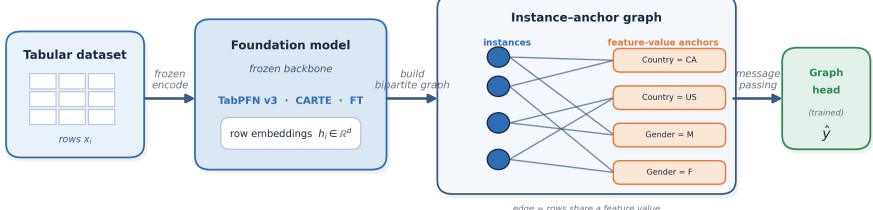

Figure 1. LATTICE. A pretrained TFM produces frozen row embeddings. A bipartite graph connects each row to anchor nodes—its categorical values and numeric bins—so two rows interact only when they share local feature structure. A small graph transformer refines the row embeddings before a linear classifier.

the lift

$$\Delta(D) \;=\; \mathrm{F1}_{+\mathrm{graph}}(D) - \mathrm{F1}_{\mathrm{native}}(D),$$

the change in macro-F1 from adding the graph. Because $\Delta$ is heavy-tailed, we test it with the two-sided Wilcoxon signed-rank test (which tests the median) and report DerSimonian–Laird random-effects pooled effects in macro-F1 points (Demšar, 2006; DerSimonian & Laird, 1986).

## 5. Results

### 5.1. The benefit tracks encoder weakness

Table 1 reports the lift for each backbone, ordered by native macro-F1. The pattern is monotone: the graph helps the weakest encoder the most and the strongest encoder the least. For CARTE it adds +7.8 points on average (median +2.8; significant at $p = 4\times10^{-47}$; helping on 83% of datasets); for FT-Transformer it adds +2.4 points; for TabPFN it adds essentially nothing on average ($-0.1$), with a negative median and a win rate of only 35%. The Wilcoxon test is significant for TabPFN as well, but in the negative direction—on the median dataset the graph slightly hurts the strongest encoder.

A single mechanism explains the trend. The graph pulls every encoder toward a common ceiling near 0.77 macro-F1: it lifts CARTE from 0.677 to 0.755 and FT from 0.739 to 0.763, but TabPFN already sits at that ceiling (0.771), so there is nothing left to add. The graph is an equalizer, not a universal improvement.

### 5.2. No single method dominates

Table 2 places all eight method variants on the 413-dataset complete-case set. Native and graph-augmented TabPFN are tied at the top: the graph raises the mean slightly (0.776 vs. 0.774) but leaves TabPFN-native ahead on median, rank, and per-dataset wins (154/413 vs. 53), and no other variant

Table 1. Graph benefit by backbone, ordered by encoder strength. Lift = (+graph) − native macro-F1 per dataset (averaged over seeds). Win% is the fraction of datasets with positive lift; $p$ is the two-sided Wilcoxon signed-rank test on the lift.

| Backbone | $n$ | Native | +Graph | Med | Win% | $p$ |
|---|---|---|---|---|---|---|
| CARTE | 418 | 0.677 | 0.755 | +.028 | 83% | 4e-47 |
| FT | 432 | 0.739 | 0.763 | +.001 | 59% | 2e-8 |
| TabPFN | 448 | 0.771 | 0.770 | −.004 | 35% | 3e-2 |

Table 2. Common-set leaderboard ($n = 413$). Mean/median macro-F1, average rank (lower is better), and number of datasets on which each variant is best.

| # | Method | Mean | Median | Rank | #best |
|---|---|---|---|---|---|
| 1 | TabPFN + graph | 0.776 | 0.821 | 3.72 | 53 |
| 2 | TabPFN native | 0.774 | 0.831 | 3.25 | 154 |
| 3 | FT + graph | 0.766 | 0.807 | 4.24 | 23 |
| 4 | CatBoost | 0.764 | 0.806 | 3.96 | 29 |
| 5 | XGBoost | 0.760 | 0.810 | 4.27 | 33 |
| 6 | CARTE + graph | 0.756 | 0.793 | 4.91 | 29 |
| 7 | FT native | 0.743 | 0.778 | 4.73 | 31 |
| 8 | CARTE native | 0.678 | 0.696 | 6.93 | 5 |

beats TabPFN. The headline, though, is that even the winner is best on only 37% of datasets—on the remaining 63% some other method wins, precisely the opening a per-dataset rule can exploit (Section 5.4).

### 5.3. Where the graph helps or hurts is predictable

The average effect on TabPFN is near zero, but this hides a strong, learnable structure. For each dataset we extract meta-features with PyMFE (Alcobaça et al., 2020), correlate them with the lift, and search for simple conjunctive rules (thresholds at data quantiles $p30/p50/p70$) whose lift is significant by the Wilcoxon test. Table 3 lists the sharpest rules. One meta-feature dominates: can_cor, the canonical correlation between features and label—how linearly predictable the label already is. When the label is hard to predict linearly (can_cor $\leq p30$) the graph helps; the

Table 3. Sharpest regimes for TabPFN. Median lift and Wilcoxon $p$ within meta-feature rules. cc (can_cor): label–feature predictability; imb: imbalance ratio; min: minority-class fraction; ae: attribute entropy; ent: class entropy; cont: fraction continuous.

|  | Regime | $n$ | Med | $p$ |
|---|---|---|---|---|
| Helps | cc$\leq p30$ | 133 | +.019 | 5e-6 |
|  | imb$\geq p70$, cc$\leq p30$ | 53 | +.066 | 2e-7 |
|  | min$\leq p30$, ae$\geq p70$, cc$\leq p30$ | 16 | +.114 | 1e-3 |
| Hurts | cont$\geq p70$ | 136 | −.007 | 3e-5 |
|  | ent$\geq p50$, ae$\leq p50$, cont$\geq p70$ | 10 | −.162 | 4e-2 |

effect sharpens as we add class imbalance and label entropy, reaching +11.4 points on a 16-dataset 3D regime (helping on 13 of 16). Conversely, the graph hurts on continuous-heavy tables with poor categorical anchors (frac_cont $\geq p70$), down to −16 points. Intuitively, the graph helps exactly when the backbone's own signal is weak and shared feature values carry information the encoder missed.

The contrast with CARTE confirms the mechanism. Applying the same search to CARTE's lift yields zero significant "hurts" regimes in one and two dimensions (and a single marginal one in 3D), versus 31 for TabPFN. For a weak encoder the graph is almost unconditionally good; for a strong one it is a conditional bet whose outcome the meta-features reveal.

### 5.4. Prediction enables routing

The predictability of the effect turns augmentation into a per-table decision: we forecast whether the graph will help before training it, and add it only when the answer is yes. A small gradient-boosted tree model over the cheap meta-features predicts each table's lift— trained only on the other datasets, so the reported gains reflect unseen tables—and we add the graph when the predicted lift is positive. This rule needs no test labels and beats both fixed policies, always-and never-augment, on every backbone. The payoff is largest for TabPFN, where the graph helps on only 35% of datasets: the router adds it to the 45% of tables predicted to benefit, improving them by +3.8 macro-F1 points on average (+1.7 over all tables). These per-dataset gains are heavy-tailed: the graph exceeds +10 points on 47 datasets. Its value for the strongest encoder is therefore not zero but conditional.

These regimes are not artifacts of searching. Over 20 random 50/50 splits, we discover the significant rules on one half and re-test each on the held-out half, counting a rule as replicated only if it is again significant there ($p < 0.05$) with the same sign. Averaged over splits, 82% of the discovered "hurts" rules and 49% of the "helps" rules replicate—versus 2.5% expected by chance—so the conditions generalize.

### 5.5. A by-product: foundation model or GBDT?

The same regime machinery answers a related, practical question: when does a tabular FM beat a tree ensemble? Comparing TabPFN-native to CatBoost and XGBoost per dataset, TabPFN wins overall on 58% and 63% of datasets respectively—but the split is again regime-dependent. TabPFN wins on wide, data-starved tables (few features, high feature-to-row ratio) by up to 20 points; the trees win on low-signal, imbalanced, high-cardinality tables by up to 11 points. Notably, this is the same regime as our main result—trees beat TabPFN on low-signal, imbalanced tables, exactly where the graph helps it—and there augmenting TabPFN beats switching to a tree: TabPFN+graph reaches 0.63 macro-F1 versus 0.58 for the best tree, winning on 70% of datasets. Details are in Appendix A.4.

### 5.6. Efficiency

The graph augmentation is lightweight. Training the graph head takes a median of 17, 41, and 46 seconds per dataset for FT, CARTE, and TabPFN respectively—on par with tuning a gradient-boosted tree (XGBoost 14 s, CatBoost 34 s median). Cost is heavy-tailed: the largest graphs reach tens of minutes at the 95th percentile. For the median dataset, however, the augmentation adds only seconds on top of the frozen backbone, whose own inference cost it inherits and does not increase. Full percentiles are in Appendix A.5.

## 6. Conclusion

We asked when an explicit graph prior helps a tabular foundation model, and found a clear, actionable answer. The benefit is largest for weaker encoders, where the graph acts as an equalizer that lifts them toward the state-of-the-art ceiling; and even for TabPFN, whose average effect is neutral, it still delivers sizable gains on identifiable low-signal, imbalanced regimes. Crucially, where it helps is predictable from cheap dataset meta-features, turning the leaderboard question into one of model selection: rather than declaring a single average winner, one can decide per dataset whether a structural prior will help and route accordingly. We hope this encourages regime-aware evaluation—asking not just which method wins on average, but when its structural assumptions fit.

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

## A. Additional Details

This appendix collects implementation and evaluation details omitted from the main text: hyperparameter optimization, regime discovery, the full regime tables and the CARTE contrast, a foundation-model-versus-GBDT map, runtime percentiles, and limitations.

### A.1. Training and Hyperparameter Optimization

The frozen backbone provides row embeddings; only the graph head is trained, per dataset. We tune the head with Optuna over 100 trials, each trained for up to 2000 epochs with early stopping (patience 25), using Adam, gradient clipping, and mixed precision. The search space is: number of graph layers $\in [2, 4]$; hidden dimension $\in \{64, 128, 256\}$; attention heads $\in \{2, 4\}$; dropout $\in [0.1, 0.5]$; learning rate $\in [10^{-4}, 10^{-2}]$; weight decay $\in [10^{-6}, 10^{-3}]$.

To handle datasets of widely varying scale, we estimate the GPU footprint before each trial and switch from full-graph training to neighbor sampling when the estimate exceeds available memory; on out-of-memory errors we reduce the neighborhood and batch sizes and retry. XGBoost and CatBoost are tuned with 100-trial Optuna and early stopping; TabPFN v3 uses its official implementation and defaults.

### A.2. Regime Discovery Procedure

To identify when the graph helps, we compare each backbone's graph-augmented and native scores per dataset and form the lift $\Delta(D)$. We extract dataset meta-features with PyMFE (Alcobaça et al., 2020)—measures of scale, class balance, and structural complexity—plus a small set of interpretable custom features (fractions of continuous, low-cardinality, high-cardinality, and binary columns). We correlate each meta-feature with $\Delta$, keep the interpretable high-signal candidates, and search conjunctive rules formed by thresholding one, two, or three features at the $p30/p50/p70$ quantiles, keeping rules with at least ten datasets whose lift is significant under the two-sided Wilcoxon test. The single most informative predictor is can_cor, the canonical correlation between features and label.

Held-out validation. To rule out over-search, we repeat the following over 20 random 50/50 splits of the datasets into a discovery half and a held-out half. On the discovery half we enumerate all one- and two-feature threshold rules and keep the significant ones (at least 12 datasets, $|\text{median lift}| \geq 0.005$, two-sided Wilcoxon $p < 0.05$), recording each rule's sign. We then re-evaluate every discovered rule on the held-out half; it replicates iff the same rule selects at least 12 datasets there, is Wilcoxon-significant ($p < 0.05$), and has the same sign as on the discovery half. For each split we compute the fraction of discovered rules that replicate, separately for "helps" and "hurts," and average across splits. For TabPFN, 82% of "hurts" rules and 49% of "helps" rules replicate. The chance rate for a single rule is 2.5%: a two-sided test at $p < 0.05$ gives a 5% false-positive rate, halved by the requirement that the sign match.

### A.3. Full Regime Tables and the CARTE Contrast

Table 4 gives the sharpest "helps" and "hurts" regimes for TabPFN across dimensions; Table 5 gives the "helps" regimes for CARTE. The qualitative contrast is stark: TabPFN has 31 significant "hurts" regimes (the graph is a conditional bet), while CARTE has none in one or two dimensions and a single marginal one in three (the graph is almost unconditionally good). The same axis—low can_cor, i.e. weak native signal—drives "helps" in both cases.

### A.4. When to Use a Foundation Model vs. a GBDT

The same regime machinery yields a practical by-product: a map of when a tabular FM beats a tree ensemble. Comparing TabPFN-native to CatBoost and XGBoost per dataset, TabPFN wins overall on 58% (vs. CatBoost) and 63% (vs. XGBoost), but the split is regime-dependent. TabPFN wins on wide, data-starved, high-feature-count tables—by up to 20 points on a 3D regime with few features and high feature-to-row ratio—while GBDTs win on low-signal, imbalanced, high-cardinality tables—by up to 11 points. The governing axis is the feature-to-row ratio (high $\rightarrow$ FM, low $\rightarrow$ trees). This composes with the graph result: the graph helps TabPFN partly in

Table 4. TabPFN: sharpest regimes by dimension (median lift, win rate, Wilcoxon $p$).

| Dir. | Dim | Regime | $n$ | Med | $p$ |
|---|---|---|---|---|---|
| Helps | 1D | can_cor $\leq p30$ | 133 | +.019 | 5e-6 |
| Helps | 2D | minority$\leq p30$, can_cor $\leq p30$ | 48 | +.066 | 8e-7 |
| Helps | 2D | imbalance$\geq p70$, can_cor $\leq p30$ | 53 | +.066 | 2e-7 |
| Helps | 3D | minority$\leq p30$, attr_ent$\geq p70$, can_cor $\leq p30$ | 16 | +.114 | 1e-3 |
| Hurts | 1D | frac_cont$\geq p70$ | 136 | −.007 | 3e-5 |
| Hurts | 2D | attr_ent$\leq p50$, mean_card$\geq p70$ | 27 | −.091 | 8e-3 |
| Hurts | 2D | n_feat$\leq p30$, feat/rows$\geq p70$ | 28 | −.076 | 1e-2 |
| Hurts | 3D | class_ent$\geq p50$, attr_ent$\leq p50$, frac_cont$\geq p70$ | 10 | −.162 | 4e-2 |

Table 5. CARTE: sharpest "helps" regimes. There are no significant "hurts" regimes in 1D/2D and one marginal 3D regime.

| Dim | Regime | $n$ | Med | $p$ |
|---|---|---|---|---|
| 1D | can_cor $\leq p30$ | 129 | +.061 | 7e-19 |
| 1D | imbalance$\geq p70$ | 130 | +.060 | 3e-17 |
| 2D | n_feat$\leq p30$, attr_ent$\leq p30$ | 28 | +.196 | 7e-4 |
| 3D | attr_ent$\leq p30$, can_cor $\leq p30$, frac_low$\leq p70$ | 10 | +.418 | 2e-3 |

the low-can_cor/imbalanced regime where GBDTs were previously ahead.

## A.5. Runtime Details

We report the per-dataset cost of training the graph head across the three backbones. Median training times are 46 s (TabPFN), 41 s (CARTE), and 17 s (FT), with 95th percentiles of roughly 1,400–2,100 s driven by a small number of very large graphs. For comparison, median gradient-boosted-tree fit times are 14 s (XGBoost) and 34 s (CatBoost), with 95th percentiles of 180 s and 960 s. These training costs include per-dataset Optuna tuning of the graph head and could be reduced with fixed defaults. Inference cost is dominated by the frozen backbone: the graph head adds a single forward pass over the instance–anchor graph and does not change the backbone's own inference behavior. The graph augmentation is therefore a lightweight, per-dataset add-on rather than a new heavyweight model.

## A.6. Coverage and Limitations

Our study covers classification, one graph construction, and one graph-head family; other constructions, stronger backbones, or regression may shift the regime boundaries. Graph training is memory-bounded: on roughly 15–20 very large datasets ($> 50{,}000$ instances) the instance–anchor graph does not fit in host memory, so those datasets are excluded from graph training and only enter the native comparison. The largest tables are trained on subsampled rows and evaluated on full test sets. The identified regimes are selective, and their prevalence is shaped by current public benchmarks, which skew small; in domains with high instance-to-feature ratios (e.g. telecom, healthcare, finance) the favorable regimes may be more common than their benchmark frequency suggests. Finally, the transductive setting assumes access to unlabeled test rows at graph-construction time, which may not hold in fully online deployment.

