# OpenReview forum: "Beyond Average Leaderboards: When Explicit Graph Priors Help Tabular Foundation Models"
_ICML.cc/2026/Workshop/FMSD — FMSD @ ICML 2026 Poster_

### Official Review · Reviewer_9uza · 2026-05-20
**LATTICE: bipartite graph on top of TP-BERTa for classification**

**Rating:** 5
**Confidence:** 5

**Review:**

This paper introduces LATTICE, a two-step tabular classification algorithm. The architecture has two components:
- A frozen backbone embedding each row into a vector in a fixed hidden dimension; TP-BERTa is chosen here.
- A bipartite graph, with nodes representing either rows (initialized with the above backbone) or anchors (initialized with learnable embeddings).
In this setup, an "anchor" is either a unique value (of a categorical column) or a bin value (of a numeric column). The graph connects row-nodes to anchor-nodes if and only if that value/bin appears in the corresponding row. For small tables, additional anchor-to-anchor connections are added (so, not really bipartite).

The resulting model is presented as a foundation model, although there appears to be no pretraining (except that of TP-BERTa). Instead, the graph is trained on each downstream task. Evaluation is performed on the union of several benchmarks from various papers (TP-BERTa, TabArena, TabZilla, PMLB) for a total of 454 tables (of which 413 are supported by all baselines). Macro-F1 score is used to evaluate against four baselines (XGBoost, CatBoost, MITRA, and TabPFN) in multiple variants. In the result table, LATTICE (in transductive mode: so unlabelled test points influence each other) surpasses all baselines except TabPFN-2.5. A subset of tables is identified where it beats this baseline as well.

The main strength of the paper is introducing a new architecture. The idea of a frozen backbone followed by per-dataset fine-tuning might be exploitable, although the results presented here are not particularly strong.

However, there are several areas of improvement, especially if this paper is intended to serve as the basis for a full conference paper. See the comments below.
- The presentation would need additional care across the whole spectrum: linguistically (many repetitions—for example, the "central" or "relevant" or "practical" question that "remains unclear" being listed five times in half a page of introduction; a part of the abstract repeated verbatim in the introduction; etc.), aesthetically (the main figure appears to be a low-resolution raster image with spell-check highlight rather than a vector graphics export), and most importantly in substance. Details are only sparsely presented, and most of the architecture description is either absent or marginally discussed in the appendix. While some brevity is understandable in four pages, the abundant repetitions suggest that this brevity was not motivated by space constraints.
- The architecture of LATTICE is slightly perplexing. No two rows are connected with fewer than two hops (i.e., going through an anchor). However, the graph model itself is very shallow (the main section of the paper mentions a single transformer layer; the appendix seems to discuss 2 or 4 of them, chosen through HPO), strongly limiting the amount of message passing that can occur. Furthermore, depending on the (undisclosed) number of bins used for numeric values, the resulting graph might end up being disconnected at random blocks, which prevents message passing even more. All in all, it seems hard to understand how such a model can effectively solve a tabular learning problem. Since the models are fine-tuned on each task, message passing is not strictly needed, but in that case, the graph structure itself might be unjustified, and a simpler decoder could be used.
- It would be interesting to see if results change when switching from TP-BERTa to a more modern row encoder, such as the one from TabICL (v2).
- In fact, it would be interesting to have TabICL v2 as a baseline as well, as it is now one of the leading open-source tabular foundation models.
- Since LATTICE builds on TP-BERTa, would it be possible to apply TP-BERTa directly on these tasks and compare the results? This also appears to be missing.
- The main result table is presented for the transductive version of LATTICE. Transductive learning is a valid but not very common use case for classification tasks. As such, none of the evaluated baselines is built for this, which might explain a large part of the gap between LATTICE and the baselines (indeed, inductive LATTICE trails behind).
- In general, the evaluation section needs to be improved. It is unclear which of the results are an artifact of the chosen metric (macro-F1, for which minority classes tend to drive the average when present). This is especially true because the "Coverage" and "High-effect" regimes specifically appear to single out datasets with high "minority_frac"—this might change with a different metric. Presenting results separately for different benchmarks (e.g., TabArena only), at least in the appendix, would make the picture clearer than mixing up to 454 tables of heterogeneous nature together. It is also puzzling that this study appears to identify a combined gap of around 8% between TabPFN v2 and TabPFN v2.5. While the latter is certainly superior, an 8% lead in average macro-F1 score is massive and contrasts with the previous literature. Separate results for more standard benchmarks would help frame these results more clearly.
- Related to the point above: the identification of these "Coverage" and "High-effect" regimes, which is perhaps framed as the main contribution of the paper, is in my opinion also its main weakness. Taking a large number of tables, splitting them according to up to 181 features, and running automated algorithms to identify where one of the five methods outperforms the others is almost bound to find something. Even though in that region one obtains striking-looking p-values, these are _not_ statistically sound p-values, so they should not be presented as such. A statistical study is only valid if the hypothesis is formulated _before_ looking at the results and the p-value is computed afterwards.

For the reasons above, I am proposing a score of "5: Marginally below acceptance threshold." The idea in the paper might be interesting and suitable for the workshop. The topic fits, and the scope of the paper is more than sufficient for a workshop, even though results are not state of the art. The main problem that prevents recommending acceptance is the presentation. It is not clear what exactly the model does, and whether the pieces built on top of TP-BERTa contribute anything important at all. The statistical justification for the results is flawed, and although some "broad interpretation" of p-values beyond their real statistical meaning is standard practice in the deep learning community, using them in a blanket automated search as done in this paper goes one step further, potentially invalidating them entirely.

---

### Official Review · Reviewer_bSoF · 2026-05-20
**Review of ICML 2026 Workshop FMSD Submission2.**

**Rating:** 3
**Confidence:** 5

**Review:**

- Summary: The paper explores the concept of graph priors for tabular learning. It applies few concepts present of graphs by exploiting graphical relations between row-instances and feature values. The experiments to explore when these graph priors help was carried out on numerous public benchmarks.
- Strengths: The concept of constructing graphs of row-instance and feature values is not new, but the approach to benefit existing tabular foundation models is interesting.
- Areas for Improvement: The English writing can be improved significantly. Quality of figure 1 should be improved significantly. The experiments should be explained more clearly.
- Detailed Comments: The paper rather lacks on many of the details in outlining the concept of graph-prior and the experiments. The figures seems to be unclear (both in terms of quality and explanation) and in the experiments the paper introduces terms that are rather difficult to follow (eg., coverage regime, high-effect regime). The paper also needs to clearly explain how the pre-training is done (as it states in the introduction that it does pre-training) and how the performances are evaluated. Furthermore, the inductive settings seems very unrealistic and a unfair comparison to supervised models (if not clearly justified).
- Justification of Score: Overall, the paper is lacking in clarity and is difficult to follow on the proposed method. Experiments show improvements, but difficult to grasp the justifications.

---

### Official Review · Reviewer_MiZj · 2026-05-22
**Interesting Direction for Understanding Graph Priors, but Meta-Feature Generalization Remains Unclear**

**Rating:** 6
**Confidence:** 3

**Review:**

## Summary

The paper proposes LATTICE, which augments tabular foundation model representations with an instance-anchor graph, and shows that graph-based modeling can be beneficial in some dataset regimes. The paper also proposes two dataset-level meta-features to help understand when graph priors are useful.

## Strengths

The paper raises an interesting and useful question on when do graph priors help tabular foundation models. The discovered dataset-level meta-features provide a promising starting point. Across several benchmarks, the paper shows that these meta-features may help explain when graph-based modeling is useful, making the comparison between graph-based and non-graph-based methods more interpretable.

## Areas for Improvement

My main concern is the actionability and generality of the proposed meta-feature analysis. The paper identifies two meta-features associated with when graph priors help, but it is unclear whether these features generalize to other benchmark collections or are sufficient to guide model choice on new datasets.

Relatedly, the paper should clarify how these meta-feature values should be computed in practice. If their computation is unstable, dataset-specific, or requires substantial preprocessing, then practitioners may still need to train both graph-based and non-graph-based methods and choose based on validation performance. In that case, the practical value of the meta-feature rule would be limited.

The paper would also be stronger if it provided more controlled evidence that changes in these meta-features actually affect the relative performance of graph-based and non-graph-based models, rather than only correlating with performance differences across heterogeneous datasets.

## Detailed Comments

The authors should validate whether the proposed meta-feature regime generalizes to additional benchmark suites or held-out benchmark families. This would make the regime analysis more convincing and less likely to be a post-hoc explanation of the current benchmark.

It would be useful to explain exactly how the meta-feature values can be computed and whether they are stable across reasonable preprocessing choices. This is important if the goal is to use them for practical model selection.

I suggest adding controlled dataset-transformation experiments. For example, starting from a fixed real dataset, the authors could vary the number of training samples, add noisy or redundant features, or artificially change class imbalance. This would preserve much of the original prediction logic while changing the meta-features used to explain when graph priors help. Such experiments would help determine whether the identified meta-features are genuinely predictive of the graph-based method’s advantage, rather than being post-hoc correlations across different datasets.

It would also be useful to include a comparison using the same TabPFN/TabPFN-like backbone without the graph component. This would better isolate the effect of the graph module from differences due to the backbone or training setup.

## Justification of Score

I would rate this paper as a 6. The paper raises an important and workshop-relevant point, and the proposed graph-based method and regime analysis are interesting. However, I am not fully convinced that the identified meta-features are generalizable or actionable enough for model selection on new datasets. Stronger validation of the regime analysis and more controlled experiments would make the contribution more convincing.